# Predictive factors of distant metastasis in surgically treated HPV-positive tonsil cancer

**Hyun-il Shin, Kwang-Jae Cho, Min-Sik Kim, Young-Hoon Joo**  *

Department of Otorhinolaryngology, The Catholic University of Korea, College of Medicine, Seoul, Korea

* joodoct@catholic.ac.kr

## Abstract

### Background

Human papilloma virus (HPV)-related tonsil cancer is associated with favorable outcomes.

### Objective

The purpose of this study was to define factors affecting distant metastasis in patients with surgically treated HPV-positive tonsil cancer.

### Methods

The present study enrolled 76 patients diagnosed with HPV-positive tonsil cancer who underwent primary surgery between January 2010 and December 2021.

### Results

Twelve (15.7%) patients experienced a distant failure with a median follow-up time of 43 months. Sites of distant metastasis included the lung (n = 10), liver (n = 1), and brain (n = 1). Upon multivariate analysis, an advanced T stage (odds ratio [OR]: 13.94, 95% confidence interval [CI]: 1.29–149.863, $p$ = 0.003) and margin involvement (OR: 5.96, 95% CI: 1.33–26.76, $p$ = 0.02) were independent predictors of distant metastases. The five-year disease-specific survival for the entire cohort was 85%. The multivariate analysis confirmed that distant metastasis (hazard ratio [HR]: 12.688, 95% CI: 3.424–47.016; $p$ < 0.001) and margin involvement (HR: 6.243; 95% CI: 1.681–23.191; $p$ = 0.006) were significant factors associated with the five-year disease-specific survival.

### Conclusion

HPV-positive tonsil cancer patients with an advanced T stage and a positive surgical margin have a substantial risk of distant metastases. Distant metastasis and margin involvement are factors that affect their survival.

**Data Availability Statement:** Data cannot be shared publicly because data contain potentially identifying or sensitive patient information. Data are available from the Seoul St Mary's Hospital (Seoul) Data Access / Ethics Committee (The

Institutional Review Board of Seoul St Mary's Hospital, https://eirb.cmcnu.or.kr/irb.do) for researchers who meet the criteria for access to confidential data.

**Funding:** This work was supported by the Korea Medical Device Development Fund grant funded by the Korea government (the Ministry of Science and ICT, the Ministry of Trade, Industry and Energy, the Ministry of Health & Welfare, the Ministry of Food and Drug Safety) (Project Number: 202011D15). The funders had no role in study design, data collection and analysis, decision to publish, or preparation of the manuscript.

**Competing interests:** The authors have declared that no competing interests exist.

# Background

Human papilloma virus (HPV)-associated oropharyngeal cancer (OPC) occurs through the carcinogenesis of p53 degradation and retinoblastoma inactivation. Among HPV types, HPV type 16 is the main cause of this cancer, and it is characterized by high levels of the p16 marker [1]. In the head and neck region, the main sites associated with HPV are those composed of lymphoid tissues, such as the tonsil and the base of the tongue [2]. Since 1970, HPV-associated head and neck cancer has been increasing more rapidly in men [2, 3]. The average age at diagnosis is 55–60 years, which is about 10 years younger than the mean age at diagnosis of HPV-unrelated head and neck cancer [2]. This finding may be explained by differences in men's smoking habits, sexual behavior, and sexual organ structure as well as changes in trends of sexual behaviors (more oral sexual partners or oral sex at an earlier age in recent than in past generations) combined with fewer tobacco-related cancers in young cohorts, which have made the outcomes of HPV-positive cancers more apparent. In addition, HPV's increased prevalence in the cervix rather than the penis might raise the chances of HPV infection in those who perform oral sex on women, contributing to the higher rate of HPV-associated oropharyngeal cancer in men [2]. In view of the importance of tobacco use in HPV-related OPC, some studies have reported that oral HPV infection is common in both smokers and non-smokers, and it is an important cause of OPC in both groups [4, 5]. Although a higher proportion of individuals with HPV-positive compared with HPV-negative tumors are non-smokers or neither smoke nor drink alcohol, many with HPV-positive disease have a history of alcohol and tobacco use [4, 5]. This finding underscores that HPV-associated OPC not only arises in people who do not smoke or drink alcohol but also occurs in people with the more traditional risk factors of tobacco and alcohol use. According to a big data study from 2007, the smoking rate in the United States peaked in the 1960s and has been on a downward trend since then [6]. In addition, among head and neck cancers, the prevalence of laryngeal cancer, which is most affected by smoking, is decreasing, and the decrease in the prevalence of OPC is less than that of laryngeal cancer [6]. This trend suggests that the proportion of HPV has increased. In a study of the oncologic outcome of OPC according to HPV status that was published in 2021, for those who were p16-positive, the five-year survival rate was about 80%, and the five-year disease-free survival was about 85% [7]. However, for those who were p16-negative, the five-year survival rate was about 40%, and the five-year disease free survival rate was about 60%. About 15% of patients with p16-positive OPC experience a recurrence within five years of diagnosis, especially in the case of tonsillar cancer; its recurrence is quite high, reaching 35% for distant metastasis [7]. Several institutions have found that the incidence of HPV-positive OPC of distant metastasis does not appear to be lower than that of HPV-negative disease and that it is now likely to be a major cause of death in these patients [8–10]. Various risk factors have been identified in a small number of studies that predicted higher distant recurrence rates, including T4 stage disease, N2c nodal stage, and extensive smoking exposure [8–10]. In a study by Weller et al., T4 disease, actively smoking, and the use of cetuximab in place of cytotoxic chemotherapy significantly affected the risk of distant failure and resulted in distant metastasis rates over 20% [11]. However, the current literature does not provide sufficient insight into the predictive factors of post-operative distant metastasis in patients with HPV-positive OPC without neoadjuvant therapy. Thus, the objective of this study was to analyze the outcomes of our HPV-positive tonsil cancer patients to identify factors associated with high risks of distant metastases to develop a proper curative plan.

## Materials and methods

### Ethics statement

The Institutional Review Board of Seoul St Mary's Hospital (Seoul) approved this retrospective review of medical records and the use of archived tumor specimens (KC21RASI0406). Due to its retrospective nature without individually identifiable or sensitive information, the requirement for informed consent was waived.

### Study population

A retrospective chart review of 148 patients diagnosed with squamous cell carcinoma of the tonsils who underwent upfront surgical resection of the primary tumor with neck dissection at the Department of Otolaryngology-Head and Neck Surgery at the Catholic University of Korea (Seoul, Korea) from 2010–2021 was performed. The treatment protocol at this center is determined on the tumor board based on the National Comprehensive Cancer Network (NCCN) 2021 guidelines. The cohort included in this study included only cases where observation or adjuvant therapy was performed after surgical treatment in line with the NCCN standard protocols. Patients who satisfied at least one of the following criteria were excluded: (1) patients who received neo-adjuvant chemotherapy (n = 41); (2) patients who were HPV-negative or p16-negative (n = 23); or (3) patients having distant metastases during the initial staging (n = 8). Finally, 76 patients were included as our cohort. The patients' characteristics (including sex and age) were analyzed, and their HPV status was confirmed by *in situ* hybridization for HPV DNA or by strong and diffuse staining for p16. Patients were assigned a T classification and an N classification that corresponded to the eighth edition of the TNM classification of the American Joint Committee on Cancer (AJCC) using their HPV status. Smoking status was divided into the following three groups: (1) never smokers; (2) former smokers (defined as having quit > 3 months at the time of diagnosis); or (3) current smokers (smoking at the time of diagnosis). Excision of the tonsil site was performed via a transoral cavity or a lateral pharyngotomy approach. For patients with clinically suspected or cytologically proven positive lateral lymph nodes, modified radical neck dissection was performed for the involved side, and selective neck dissection was performed for clinical N0 patients. The postoperative surveillance protocol of our center is as follows. Chest Computed tomography (CT), abdomen-pelvis CT, bone scan, and primary site magnetic resonance imaging (MRI) were performed every six months after the end of treatment for five years. If new symptoms such as pain arose during the follow-up period, an imaging study was performed on the relevant area to check for any abnormalities. If a lesion suspected of distant metastasis was found, additional tests such as biopsy or positron emission tomography-CT were performed to confirm distant metastasis. Patients with images demonstrating evidence of distant metastatic disease were re-staged, and biopsy confirmation at the site of distant metastasis was attempted in anatomically accessible cases. The time to distant metastasis was calculated from the day of surgery. Primary site surgical specimens were carefully examined with respect to their size, depth of invasion, margin involvement (clear; close [< 5 mm]; positive [< 1 mm]), margin type (infiltrative; expanding), and lympho-vascular-neural invasion. The number, size, bilaterality, extranodal extension (ENE), and involved neck level of positive lymph nodes in neck dissection specimens were also carefully analyzed.

### Statistical analysis

Statistical analyses were performed using the Statistical Package for the Social Sciences (SPSS Inc., Chicago, IL, USA). Student's t-tests and Pearson's chi-square tests were used to analyze

the patient and tumor characteristics. A logistic analysis was carried out for the univariate and multivariate analyses. A Kaplan–Meier analysis was used for the survival analysis. A Cox proportional hazards regression was performed to determine the survival analysis. The variables examined included age, smoking status, T classification, N classification, type of adjuvant therapy, and pathology results.

## Results

### Patient characteristics and surgical parameters

The characteristics of the 76 patients of the study are summarized in Table 1. Among the 76 cases(61 men and 15 women), the average age was 68.5 years (range: 43–77 years). Fifteen patients (19.7%) had an advanced T stage (T3 or 4), and 12 patients (15.7%) had an advanced N stage (N2). Regarding the surgery type, a transoral approach (97.3%) was the dominant method used. For adjuvant therapy, 26 (34.2%) patients received radiation-only therapy, while 36 (47.4%) received concurrent chemoradiation therapy (CCRT). Regarding the surgical

**Table 1. Demographic profiles of patients with HPV positive tonsil cancer (n = 76).**

| Parameter | No of patients (%) |
|---|---|
| **Age** | 68.5 |
| **Sex (M:F)** | 61:15 |
| **Smoking** | |
| Currnet smoker | 6 (7.8%) |
| Former smoker | 19 (25%) |
| Never smoker | 51 (67.1%) |
| **pathologic T classification** | |
| T1 | 20 (26.3%) |
| T2 | 41 (53.9%) |
| T3 | 14 (18.4%) |
| T4 | 1 (1.3%) |
| **pathologic N classification** | |
| N0 | 12 (15.7%) |
| N1 | 52 (68.4%) |
| N2 | 12 (15.7%) |
| **Adjuvant therapy** | |
| None | 14 (18.4%) |
| RT only | 26 (34.2%) |
| CCRT | 36 (47.4%) |
| **Margin involvement** | |
| Clear | 56 (73.6%) |
| Close (<5mm) | 4 (5.2%) |
| Positive (<1mm) | 16 (21%) |
| **Margin type** | |
| Infiltrative | 21(27.6%) |
| Expanding | 16 (21%) |
| Both | 39 (51.3%) |
| **Lymphovascular invasion** | 34 (44.7%) |
| **Perineural invasion** | 4 (5.2%) |
| **Depth of invasion (>1cm)** | 49 (64.4%) |
| **Extranodal extension** | 21 (27.6%) |

margin, a positive margin was found in 16 (21%); 56 participants (73.6%) had a negative margin, and the remaining 4 patients (5.2%) had a close margin. Among the patients with a positive margin, 13 out of 16 (81.2%) received CCRT, 2 patients (1.6%) underwent radiation therapy, and 1 patient was observed with active surveillance due to that patient's refusal of adjuvant therapy. The margin type was infiltrative in 21 (27.6%), the expanding type in 16 (21%), and both types in 39 (51.3%) patients. Lymphovascular invasion was observed in 34 patients (44.7%), and perineural invasion was documented in 4 patients (5.2%). Forty-nine (64.4%) patients showed ≥ 1 cm depth of invasion of the primary tonsil site. Twenty-one (27.6%) patients had extranodal extension of the cervical lymph node. For margin status, 6 patients (50%) showed a close margin (between 1mm and 5mm), and 6 patients demonstrated a negative margin (≥ 5mm). An infiltrative margin type was observed in 9 patients (75%), and 3 patients (25%) showed an expanding margin type. Seven patients (58.3%) had lympho-vascular invasion, while 1 patient (8.3%) had perineural invasion. The mean depth of invasion was 15.16 mm, and ENE was observed in 4 patients (25%)(Table 1.). In the entire cohort, 12 (15.7%) patients experienced a distant failure. Of these 12 (15.7%) patients, 2 patients had both locoregional and distant failures with all-lung metastasis. The median time to distant metastases was 32 months (range: 7–96 months). Two patients showed early distant metastasis (within one year). The sites of distant metastasis included the lung (n = 10, 83.3%), liver (n = 1, 8.3%), and brain (n = 1, 8.3%). Ten (83.3%) patients had undergone treatment for their distant metastasis, including six patients who received chemotherapy alone, two patients who received chemoradiation, and two patients who received both surgery and chemotherapy. Surgical interventions included wedge resections for pulmonary metastases in two patients. Four patients (33.3%) had a smoking history, and the same, 4 patients had an alcohol use history (Table 2.).

### Risk factors for distant metastasis

A univariate analysis revealed that an advanced T stage (6.5% vs. 53.3%, $p = 0.01$), margin involvement (10% vs. 37.5%, $p = 0.01$), and margin type (infiltrative: 15% vs. expanding: 10.9%, $p = 0.04$) were significantly associated with distant metastasis. Upon multivariate analysis, an advanced T stage (odds ratio [OR]: 13.94; 95% confidence interval [CI]: 1.29–149.86; $p = 0.01$) and margin involvement (OR: 5.96; 95% CI: 1.33–26.76; $p = 0.02$) were independent predictors of distant metastases (Table 3), whereas the N classification and adjuvant therapy modality were not significantly associated with the presence of distant metastases.

### Distant metastasis and survival

The median follow-up time was 43 months (range: 2–112 months). The five-year disease specific-survival (DSS) rate and the overall survival for the entire cohort were 85% and 84.2%, respectively. The DSS rate was 79% for patients with an early T stage and 51% for those with an advanced T stage ($p = 0.001$) (Fig 1). According to the margin status, the DSS rate was 86% for a negative margin and 0% for a positive margin ($p = 0.001$) (Fig 1). In addition, the DSS rate was 87% for those without distant metastasis and 17% for those with distant metastasis ($p < 0.001$) (Fig 1). The DSS rate of the infiltrative type was 60.5%, while that of the expanding type was 83.6% ($p = 0.14$) (Fig 2). Upon multivariate analysis, distant metastasis (HR: 12.68, 95% CI: 3.424–47.016, $p < 0.001$) and margin involvement (HR: 6.24, 95% CI: 1.681–23.191, $p = 0.006$) remained significant factors associated with the five-year DSS rate (Table 4).

**Table 2. Characteristics of patients who developed distant metastasis.**

| Patient no. | Site of recurrence | T | N | Adjuvant modality | Salvage modality | Time to distant metastasis (month) | Survival | Smoking | Alcohol |
|---|---|---|---|---|---|---|---|---|---|
| 1 | Lung | 2 | 1 | CCRT | Chemo | 18 | No | No | No |
| 2 | Lung | 3 | 0 | None | Surgery, Chemo | 38 | No | No | No |
| 3 | Lung | 3 | 1 | CCRT | Chemo | 35 | No | Yes | Yes |
| 4 | Liver | 3 | 2 | CCRT | Chemo | 11 | YES | Yes | Yes |
| 5 | Lung | 1 | 1 | RT | Refuse | 96 | No | Yes | Yes |
| 6 | Lung | 2 | 2 | CCRT | Chemo, Radiation | 46 | No | No | No |
| 7 | Lung | 2 | 1 | RT | Surgery, Chemo | 61 | No | No | No |
| 8 | Brain | 3 | 1 | RT | Refuse | 58 | No | Yes | Yes |
| 9 | Lung | 3 | 2 | CCRT | Chemo, Radiation | 30 | No | No | No |
| 10 | Lung | 3 | 2 | CCRT | Chemo | 19 | No | No | No |
| 11 | Lung | 2 | 1 | RT | Chemo | 22 | YES | No | No |
| 12 | Lung | 3 | 1 | RT | Chemo | 13 | YES | No | No |

| Patient no. | Surgical margin status | Lymphatic invasion | Vascular invasion | Perineural invasion | DOI (mm) | ENE | Margin type |
|---|---|---|---|---|---|---|---|
| 1 | Close | No | No | No | 20 | No | Infiltrative |
| 2 | Close | No | No | No | 17 | Yes | Infiltrative |
| 3 | Negative | Yes | No | No | 25 | Yes | Infiltrative |
| 4 | Close | Yes | No | No | 21 | No | Expanding |
| 5 | Negative | Yes | No | No | 16 | No | Expanding |
| 6 | Negative | No | No | No | 11 | No | Both |
| 7 | Negative | Yes | Yes | No | 8 | No | Both |
| 8 | Negative | No | No | No | 20 | No | Infiltrative |
| 9 | Close | Yes | No | No | 14 | Yes | Infiltrative |
| 10 | Close | Yes | No | Yes | 16 | No | Expanding |
| 11 | Close | No | No | No | 13 | No | Infiltrative |
| 12 | Negative | Yes | Yes | No | 1 | Yes | Both |

DOI: Depth of invasion, ENE: Extra nodal extension

CCRT: Concurrent-chemoradiation therapy, RT: Radiation therapy

## Discussion

Distant metastasis has been underestimated in head and neck squamous cell carcinoma because loco-regional recurrence plays an important role in morbidity. However, about 15% of patients with p16-positive OPC will experience a recurrence within five years of their diagnosis, especially in the case of tonsillar cancer, which has a high rate (35%) of distant metastasis [7]. Ang et al. reported no difference in the rate of distant metastasis despite significant

**Table 3. Multivariate analysis associated with distant metastases.**

| Parameters | P value | OR | 95% CI |
|---|---|---|---|
| T stage | 0.03* | 13.94 | 1.29–149.863 |
| N stage | 0.47 | 4.76 | 0.06–351.96 |
| Margin involvement | 0.02* | 5.96 | 1.33–26.76 |
| Margin type | 0.08 | 1.59 | 0.07–35.04 |
| Extra nodal extension (ENE) | 0.69 | 0.49 | 0.01–16.58 |
| Adjuvant Chemo | 0.14 | 0.1 | 0.005–2.12 |
| Adjuvant Radiation | 0.17 | 15.01 | 0.299–754.45 |

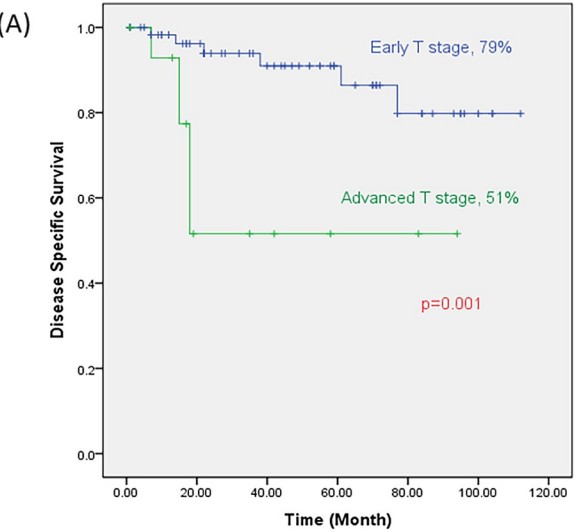

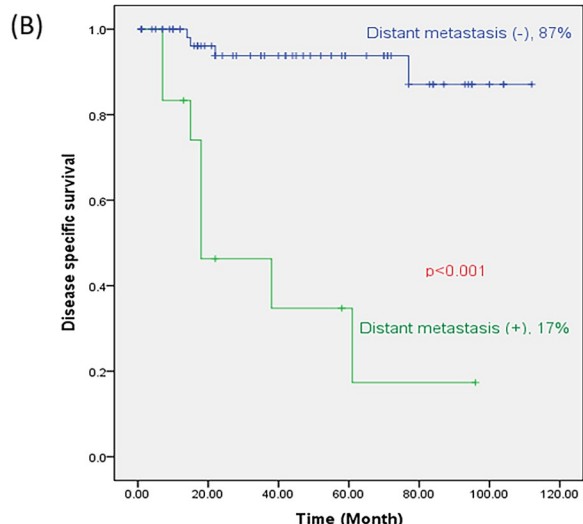

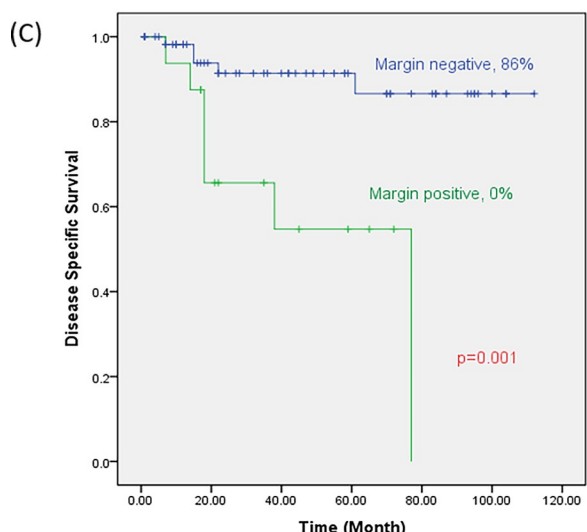

**Fig 1. Kaplan–Meier figures of disease-specific survival in patients with HPV positive tonsil cancer.** (A) according to T stage. (B) according to distant metastasis. (C) according to margin status.

improvements in local control and survival of patients with HPV-positive OPC compared to patients with HPV-negative OPC [12]. In general, in young and healthy patients, other risks of death are gradually minimized, and the impact of distant metastasis on survival becomes increasingly important [7, 11]. Therefore, it is important to better understand the predictors of distant metastasis because they might affect clinical trial design and ultimately influence treatment decisions. Most prior studies have reported the recurrence or survival of the oropharynx as a unit but not of each sub-site. However Wendt presented the recurrence and survival rates according to the sub-site of HPV-positive OPC. In his study, the primary sites consisted of the tonsil (63%), base of the tongue (22%), and other oropharynx locations (15%). In his study, in the case of the tonsil, 35% of patients experienced distant failure, which is a result of examining the distant failure rate regardless of the chosen treatment modality [7]. The treatment protocol at this center is determined on the tumor board based on the NCCN guidelines. The cohort included in this study was composed only of cases where observation or adjuvant therapy was

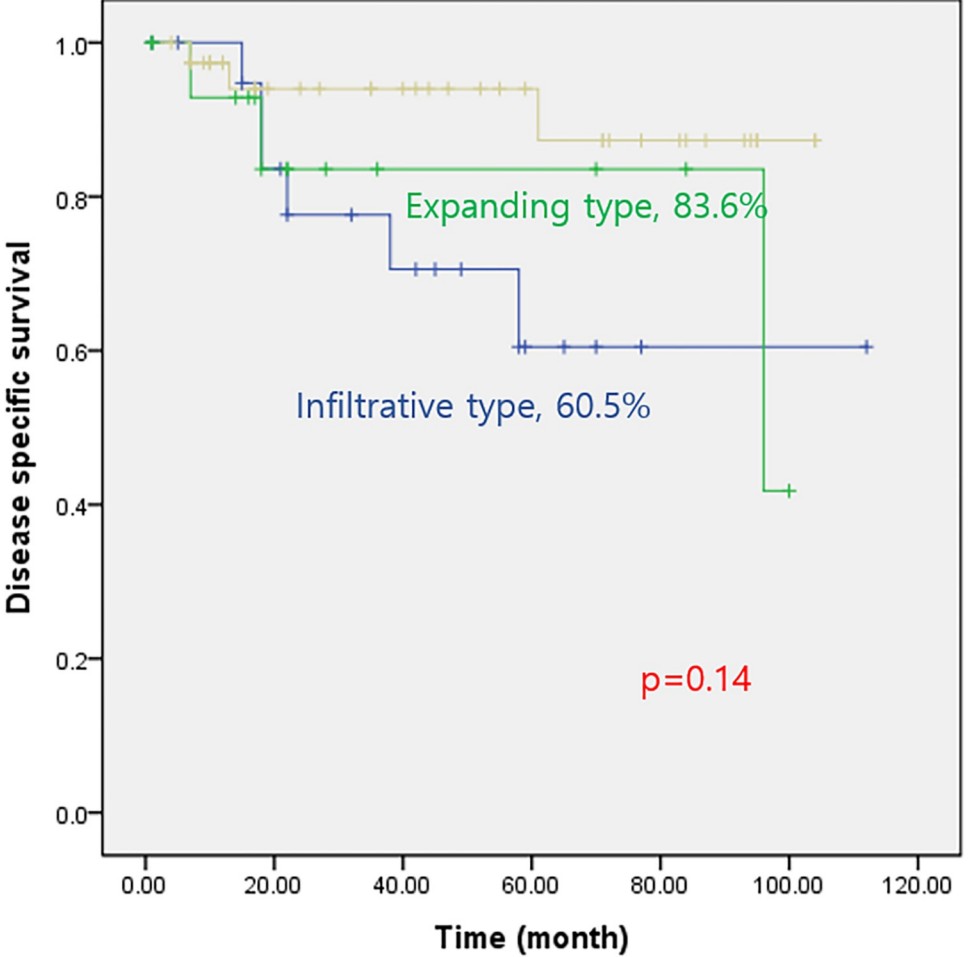

**Fig 2. Kaplan–Meier figures of disease-specific survival in patients with HPV positive tonsil cancer according to margin type.**

**Table 4. Cox multivariate analysis for survival.**

| Parameters | β (SE) | *p*-Value | Exp (β) | 95% CI of Exp (β) | |
|---|---|---|---|---|---|
| | | | | Lower | Upper |
| T stage | 1.918 | 0.24 | 6.808 | 0.271 | 171.168 |
| Distant metastasis | 2.541 | <0.001* | 12.688 | 3.424 | 47.016 |
| Margin involvement | 1.832 | 0.006* | 6.243 | 1.681 | 23.191 |
| Margin type | -0.565 | 0.55 | 0.568 | 0.081 | 4.005 |

performed after surgical treatment in accordance with the NCCN standard protocols. In the case of base of tongue lesions, our center recommends neoadjuvant chemotherapy or CCRT rather than an initial surgery with the intent to improve the patients' swallowing functional outcome. Therefore, patients with base of tongue lesions were not included in this study. Since most patients with HPV-related tonsil cancer are relatively young and healthy, it is important to limit treatment complications and toxicity. Yeh's systematic review from 2015 suggested that the rate of upfront surgery was high (87%) for early stage HPV-positive tonsil cancer [13]. In that paper, transoral robotic surgery (TORS) showed excellent results for factors related to functional outcomes such as feeding tubes and fenestration [13]. In the present study, only cases that underwent upfront surgery without induction chemotherapy were included. Distant metastasis was observed in 15% of cases, and it had an effect on survival both in the Kaplan–Meier estimate and the Cox multivariate analysis. Traditionally, tumor characteristics (such as their T and N classifications) have been the main factors of prognosis that direct staging and entry criteria determination for most clinical trials. In this study, an advanced T stage, margin involvement, and margin type (infiltrative type) were related to the occurrence of distant metastasis. In a paper published in 2019, Philippe claimed that there was a significant correlation between an advanced T stage of tonsil cancer and a positive margin [14]. Such a correlation was also expected to be found in the present study. According to another publication on a group of patients who underwent chemotherapy, the use of cetuximab in place of cytotoxic chemotherapy also resulted in a rate of distant metastases that was ≥ 20% [11]. Taken together, it can be inferred that the initial T stage observed prior to surgery or chemotherapy is an important factor that affect distant metastasis. To better understand this association, a study that compares the incidence of distant metastasis in a group of patients who have undergone surgery after receiving induction chemotherapy in the same medical center will be needed. In addition, the usefulness of delicate techniques for transoral surgery that can safely secure the margin must be confirmed. Philippe reviewed the difference in margin results according to the conventional transoral excision method (n = 301), the transoral micro laser surgery method (n = 1,012), and the TORS method (n = 1,676). Regardless of the procedure, there were no differences in results among transoral micro laser surgery, conventional transoral excision, and TORS approaches when comparing the series of margins with the use of frozen sections [14]. Therefore, it is important to select the optimal surgical method that can secure the margin in consideration of the unique conditions of the operator and patient. According to the National Comprehensive Cancer Network guidelines (2021), extra-nodal extension and margin involvement are indications for adjuvant chemoradiotherapy [15, 16]. The surgical margin may be defined as a close margin (< 5 mm) and or positive (< 1 mm). In this study, margin involvement was found to affect both distant metastasis and survival. In the case of a positive surgical margin, it was confirmed once again that adjuvant chemoradiation was necessary. In this study, the margin type was also analyzed. The margin type could be infiltrative, expanding, or both. On univariate analysis (but not on multivariate analysis), the infiltrative margin type was found to be a significant factor that affected distant metastasis. The authors could not find any

studies that analyzed the margin type (infiltrative vs. expanding) in HPV-related/non-related OPC. However, studies on the margin type of malignancy at other sites, such as the thyroid or in, colorectal cancer, do exist. Kim et al. reported the clinical outcomes of papillary thyroid carcinoma according to the tumor margin type. Kim found that an infiltrative type margin was associated with significantly higher rates of tumor multiplicity, extrathyroidal extension, coexistence of lymphocytic thyroiditis, and metastasis to the lateral neck lymph nodes than an expanding type margin [17]. Garcia-Solano et al. published a paper on pathologic features (such as tumor budding and invasive margin type) in colorectal adenocarcinoma. They found that the infiltrative margin type was a worse prognostic factor for five-year survival than the expanding type in serrated adenocarcinoma [18]. Based on these studies, an infiltrative margin type may act as a predictor of a poor oncologic outcome, which is consistent with the results of this study.

Choi et al. published a study on the prognosis according to lymph node ratio after upfront surgery in HPV-positive OPC patients, and presented a positive margin and ENE as the main indications for adjuvant therapy [19]. In her study, a positive margin was found in 25 (27.7%) out of 90 patients, and adjuvant RT was performed in 26 patients (28.9%), and CCRT was performed in 47 patients (52.2%). Roden et al. published a study comparing the outcomes of upfront surgery with adjuvant therapy or CCRT in advanced-stage HPV-positive tonsil cancer patients using a national database [20]. Surgery was performed in 8768 patients, and positive margins were observed in 2762 (31.5%). Among the patients with a positive margin, adjuvant RT was performed in 25.9% and CCRT was administered in 35.2%. In the present study, the proportion of patients who showed a positive margin was smaller than in the other two studies, and the proportion of patients who underwent CCRT was higher than in the other two studies. This findings was likely influenced by the preference of this center's tumor board. An additional point was that adjuvant therapy was found to have no significant effect on distant metastasis. However, a Cox multivariate analysis that included distant metastasis indicated that margin involvement and distant metastasis showed a tendency to adversely affect survival. It can therefore be inferred that adjuvant radiation therapy may be an effective method for increasing survival by loco-regional control. Gillison et al. analyzed patients with p16-positive HPV OPC enrolled in Phase 3 trials (RTOG 9003, 0129) and concluded that p16 status and smoking were associated with disease progression and mortality [21]. However, in the present study, there was no association between distant metastasis and tobacco use. This findings supports the previous study's conclusion that HPV-positive tonsil cancer is more associated with sexual behavior than smoking. Studies on the relationship between the metastasis site and survival have also been conducted. Finley et al. investigated patients with metastatic head and neck squamous cell carcinoma who were treated between 1970 and 1989 and found that pulmonary metastatectomy, solitary lung lesions, loco-regional control, early stage, and time-to-distant metastasis were factors associated with improvements in overall survival [22]. In the present study, 12 cases of distant metastases were found, of which 10 cases occurred in the lung, 1 case occurred in the brain, and 1 case occurred in the liver. Metastatectomy was performed for the 2 lung cases, while chemotherapy or CCRT was performed for the remaining 8 lung cases. Only 1 case treated with metastatectomy survived more than five years. Strategies for controlling distant metastasis of HPV-positive tonsil cancer should be analyzed with a larger amount of data in the future.

In conclusion, the current study is the largest and the most robust analysis to date to identify specific prognostic factors in patients with HPV-positive tonsil cancer treated with primary surgery. HPV-positive tonsil cancer with an advanced T stage (T3, T4), a positive surgical margin, and an infiltrative margin type has a substantial risk for distant metastases. Distant metastasis and margin involvement were also significantly related to survival. Such information can be used in patient counseling and to determine appropriate risk stratification.

## Author Contributions

**Conceptualization:** Young-Hoon Joo.

**Data curation:** Hyun-il Shin, Kwang-Jae Cho, Min-Sik Kim, Young-Hoon Joo.

**Formal analysis:** Hyun-il Shin, Min-Sik Kim, Young-Hoon Joo.

**Investigation:** Kwang-Jae Cho.

**Methodology:** Hyun-il Shin.

**Writing – original draft:** Hyun-il Shin, Young-Hoon Joo.

**Writing – review & editing:** Kwang-Jae Cho, Young-Hoon Joo.

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
