## [Decision Letter · Decision Letter 0]

21 Sep 2022

PONE-D-22-18441Predictive factors of distant metastasis in surgically treated HPV positive tonsil cancerPLOS ONE

Dear Dr. Joo,

Thank you for submitting your manuscript to PLOS ONE. After careful consideration, we feel that it has merit but does not fully meet PLOS ONE’s publication criteria as it currently stands. Therefore, we invite you to submit a revised version of the manuscript that addresses the points raised during the review process.

We look forward to receiving your revised manuscript.

Kind regards,

Andrew Birkeland, M.D.

Academic Editor

PLOS ONE

Journal Requirements:

 "This work was supported by the Korea Medical Device Development Fund grant funded by the Korea government (the Ministry of Science and ICT, the Ministry of Trade, Industry and Energy, the Ministry of Health & Welfare, the Ministry of Food and Drug Safety) (Project Number: 202011D15)" 

   "This work was supported by the Korea Medical Device Development Fund grant funded by the Korea government (the Ministry of Science and ICT, the Ministry of Trade, Industry and Energy, the Ministry of Health & Welfare, the Ministry of Food and Drug Safety) (Project Number: 202011D15)"

Additional Editor Comments :

We thank the authors for their submission. Please see the reviewer's comments to help improve the manuscript.

Reviewers' comments:

Reviewer's Responses to Questions

**Comments to the Author**

1. Is the manuscript technically sound, and do the data support the conclusions?

Reviewer #1: Partly

Reviewer #2: Yes

2. Has the statistical analysis been performed appropriately and rigorously? 

Reviewer #1: Yes

Reviewer #2: Yes

3. Have the authors made all data underlying the findings in their manuscript fully available?

Reviewer #1: Yes

Reviewer #2: No

4. Is the manuscript presented in an intelligible fashion and written in standard English?

Reviewer #1: No

Reviewer #2: Yes

5. Review Comments to the Author

Reviewer #1: “Predictive factors of distant metastasis in surgically treated HPV positive tonsil cancer” finds that in a retrospective review of HPV positive oropharyngeal SCC undergoing primary surgery, positive margins and advanced T stage are independent predictors of delayed distant metastases. In the same cohort, distant metastasis and margin involvement were predictive of worse survival outcomes.

This study is one of the largest reviews to date specifically looking at delayed distant metastasis in HPV mediated oropharyngeal SCC undergoing primary surgery. Factors including TNM classification, adjuvant therapy, margin involvement, margin type, and histopathologic characteristics were analyzed.

There remain some points that might be clarified:

Major:

1) The means by which patients were found to have distant metastasis remains unclear. Did all patients undergo repeat PET at certain intervals after treatment? Additional clarity on surveillance protocols would be helpful

2) For the 5 year disease specific survival, is the calculation for distant metastasis from time of treatment or from time of diagnosis of distant metastasis?

3) A table of characteristics in each patient with distant metastasis

Minor:

1) In the background, there is a statement, “This could be explained by differences in men's smoking, sexual behavior, and sexual organ structure.” Please clarify and cite

2) “Since 1970, HPVassociated head and neck cancer is increasing more rapidly in men. The average age of diagnosis is 55 to 60 years, which is about 10 years younger than HPV-associated head and neck cancer”. Clarify HPV and non-HPV cohorts

3) For patients with distant metastasis, please consider including treatments received and tumor characteristics as a separate cohort in “Patients’ characteristics and surgical parameters” section

4) Please cite any past papers looking at margin type and their conclusions

5) Were there any patients in the study that did not receive the NCCN standard of care treatment recommendations?

6) “ It can be inferred that the invasiveness of infiltrative tumor margin causing distant metastasis is stronger.” This is only significant on univariate analysis so consider not including this statement

7) “In conclusion, the current study is the largest and the most robust analysis up to date to identify”. Delete “up”

Reviewer #2: This is a retrospective review examining distant failures following primary surgical management of 76 patients with HPV-mediated tonsil cancer treated at a single institution over a 12 year period. The study aims to identify risk factors for distant failures after surgical management, which the authors note is currently lacking in the literature for patients not having undergone neoadjuvant treatment. All patients underwent primary surgical management with risk-adjusted adjuvant treatment. None had been previously treated prior to surgery. Overall survival and disease specific survival for all-comers in this cohort was excellent. Advanced T classification and positive marginal status were shown to be independent predictors of distant metastasis in this cohort by multivariate analysis. Infiltrative margin type was also associated with distant metastasis by univariate analysis only. Moreover, distant metastasis and positive marginal status were associated with significantly worsened 5-year disease-specific survival. The authors conclude that this is the largest cohort of its type, providing crucial risk stratification data and information for patient counseling in this population.

Major Points

1. The authors note that other studies have suggested that the tonsillar subsite has high rates of distant failure, but they might clarify why the tonsillar subsite was exclusively examined rather than examining both tonsillar and base of tongue subsites (or alternatively all oropharyngeal subsites). This may strengthen the study by increasing the size of the cohort and also provide additional data about the other subsites.

2. Nearly half of the patients who underwent surgery in this cohort required both adjuvant chemotherapy and radiation. It may be useful to discuss surgical selection criteria at the authors’ institution. Triple modality therapy may be unavoidable in a subset of patients with unanticipated, unresectable positive margins or unexpected extracapsular extension. However, many providers/tumor boards are likely to advocate forgoing surgery in patients who are at high risk to require triple modality therapy if subjected to surgery. While the authors report excellent overall survival comparable with other HPV-mediated cohorts, there may have been some missed opportunities to limit treatment-related morbidities in the patients undergoing triple modality treatment. How do the rates of positive margins and proportion of patients undergoing triple modality treatment in this cohort compare to those reported in other studies?

3. The effect of marginal type is reported with respect to distant metastasis. Did the authors examine the relationship of infiltrative vs expanding marginal types on survival as well?

4. Were any patients treated with immunotherapy after the diagnosis of their distant metastasis?

Minor Points:

1. (Page 9, paragraph 1, lines 2-3) P16 is a surrogate marker for HPV-mediated carcinoma rather than a “type.” The authors should clarify this.

2. (Page 9, paragraph 1, lines 5-6) the authors talk about HPV-associated cancer in men in the previous sentence and then reference diagnosis being 10 years earlier than HPV-associated cancer. Please clarify, since it seems like an error (are the authors trying to compare this to the age of diagnosis in women, for patients with HPV-negative carcinoma, or something else?)..

3. (Page 9, paragraph 1, lines 15-16) Please correct “16 negative”

4. (Page 13, paragraph 1, lines 3-4) The authors reference “advanced n-stage” as including N2 or N3. However, the AJCC 8 nodal classification system does not have a pathological N3 designation. Does this mean that the N-classification of these tumors was determined clinical evaluation alone or by combining available clinical and pathological data?

5. (Page 14, paragraph 2) when reporting the univariate analysis results, it is not always clear which variables correspond to which of the percentages in parentheses. For example, which margin type had a significantly higher association with distant metastasis?

6. The overall writing style, punctuation, and grammar should be closely examined. The authors are not consistent with indentation between various paragraphs and sections. There is also a variety of minor grammatical errors.

6. PLOS authors have the option to publish the peer review history of their article (what does this mean?). If published, this will include your full peer review and any attached files.

Reviewer #1: No

Reviewer #2: No

---

## [Author Response · Author response to Decision Letter 0]

30 Oct 2022

Reviewer #1: “Predictive factors of distant metastasis in surgically treated HPV positive tonsil cancer” finds that in a retrospective review of HPV positive oropharyngeal SCC undergoing primary surgery, positive margins and advanced T stage are independent predictors of delayed distant metastases. In the same cohort, distant metastasis and margin involvement were predictive of worse survival outcomes.

This study is one of the largest reviews to date specifically looking at delayed distant metastasis in HPV mediated oropharyngeal SCC undergoing primary surgery. Factors including TNM classification, adjuvant therapy, margin involvement, margin type, and histopathologic characteristics were analyzed.

There remain some points that might be clarified:

Major:

1) The means by which patients were found to have distant metastasis remains unclear. Did all patients undergo repeat PET at certain intervals after treatment? Additional clarity on surveillance protocols would be helpful

Response:

Thank you for your suggestion.

In our center, chest CT, abdomen-pelvis CT, bone scan, and primary site MRI were performed every 6 months after the end of treatment for 5 years. If new symptoms such as pain are found in the period of follow-up, an image study is performed on the relevant area to check for abnormalities. If a lesion suspected of distant metastasis is found, additional tests such as biopsy or PET-CT are performed to confirm distant metastasis. Patients with images demonstrating evidence of distant metastatic disease were re-staged and pathologic confirmation for site of distant metastasis had tried in available cases.

We have included this information in the Method section.

2) For the 5 year disease specific survival, is the calculation for distant metastasis from time of treatment or from time of diagnosis of distant metastasis?

Response:

Thank you for comment.

Other study sets the end point differently. McBride set the time to distant metastasis as the time from completion of surgery to the first radiographic evidence of distant metastatic disease. [McBride] In the Ann’s study, the primary end point was overall survival, defined as the time from randomization to death. Secondary end points included progression-free survival, defined as the time from randomization to death or the first documented relapse, which was categorized as local–regional disease (tumor at the primary site or regional nodes) or distant metastases.

 This study calculated the follow-up period from the day of surgery. Because, the subject of this study is that patients who have not received neoadjuvant and observed after surgery are subjects.

Ref) 

S.M. McBride et al. / Long-term survival after distant metastasis in patients with oropharyngeal cancer / Oral Oncology 50 (2014) 208–212

Ang KK, Harris J, Wheeler R, Weber R, Rosenthal DI, Nguyen-Tân PF, et al. (2010) Human papillomavirus and survival of patients with oropharyngeal cancer. N Engl J Med. 363:24-35

We have included this information in the Method section.

3) A table of characteristics in each patient with distant metastasis

Response:

Thank you for suggestion.

Of 12 (15.7%) patients who experienced a distant failure, two patients had both locoregional and distant failure with all lung metastasis. The median time to distant metastases was 32 months (range, 7–96 months). Two patients showed early distant metastasis (within 1 year). Sites of distant metastasis included lung (n = 10, 83.3%), liver (n = 1, 8.3%), and brain (n = 1, 8.3%). Ten (83.3%) patients had undergone treatment for their distant metastasis, including six patients who received chemotherapy alone, two patients who received chemoradiation, and two patients who received both surgery and chemotherapy. Surgical intervention included wedge resections for pulmonary metastases in two patients. 4 patients (33.3%) had smoking history, as same, 4 patients had alcohol history. 

For margin status, 6 patients (50%) showed close margin (1mm< <5mm) and 6 patients showed negative margin (5mm <). An infiltrative margin type was observed in 9 patients (75%) and 3 patients (25%) showed expanding margin type. 7 patients (58.3%) had lympho-vascular invasion and 1 patient (8.3%) had perineural invasion. The mean depth of invasion was 15.16 mm, and ENE was observed in 4 patients (25%).

We have included this information in the result section and supplementary table (Table 2.)

Table 2. Characteristics of patients who developed distant metastasis

Patient no. Site of recurrence T N Adjuvant modality Salvage modality Time to distant metastasis (month) Survival Smoking Alcohol

1 Lung 2 1 CCRT Chemo 18 No No No

2 Lung 3 0 None Surgery, Chemo 38 No No No

3 Lung 3 1 CCRT Chemo 35 No Yes Yes

4 Liver 3 2 CCRT Chemo 11 YES Yes Yes

5 Lung 1 1 RT Refuse 96 No Yes Yes

6 Lung 2 2 CCRT Chemo, Radiation 46 No No No

7 Lung 2 1 RT Surgery, Chemo 61 No No No

8 Brain 3 1 RT Refuse 58 No Yes Yes

9 Lung 3 2 CCRT Chemo, Radiation 30 No No No

10 Lung 3 2 CCRT Chemo 19 No No No

11 Lung 2 1 RT Chemo 22 YES No No

12 Lung 3 1 RT Chemo 13 YES No No

Characteristics of patients who developed distant metastasis (continued)

Patient no. Surgical margin status Lymphatic invasion Vascular invasion Perineural invasion DOI (mm) ENE Margin type

1 Close No No No 20 No Infiltrative

2 Close No No No 17 Yes Infiltrative

3 Negative Yes No No 25 Yes Infiltrative

4 Close Yes No No 21 No Expanding

5 Negative Yes No No 16 No Expanding

6 Negative No No No 11 No Both

7 Negative Yes Yes No 8 No Both

8 Negative No No No 20 No Infiltrative

9 Close Yes No No 14 Yes Infiltrative

10 Close Yes No Yes 16 No Expanding

11 Close No No No 13 No Infiltrative

12 Negative Yes Yes No 1 Yes Both

DOI: Depth of invasion, ENE: Extra nodal extension

CCRT: Concurrent-chemoradiation therapy, RT: Radiation therapy

Minor:

1) In the background, there is a statement, “This could be explained by differences in men's smoking, sexual behavior, and sexual organ structure.” Please clarify and cite

Response:

Thank you for your comment.

The incidence of head and neck cancers overall has declined in recent years, consistent with the decrease in tobacco use. By contrast, incidence of HPV-associated oropharyngeal cancer seems to be rising. In particular, the prevalence is increasing rapidly in young men.[Shiboski] This effect may be attributable to changes in sexual norms (more oral sex partners or oral sex at an earlier age in recent than past generations) combined with fewer tobacco-related cancers in young cohorts, making the outcomes of HPV-positive cancers more apparent.[Marur] In addition, HPV prevalence in cervix rather than penile tissue might boost the chances of HPV infection when performing oral sex on a woman, contributing to the higher rate of HPV-associated oropharyngeal cancer in men.[Marur] In view of the importance of tobacco use in HPV related OPC, there are some study that reported oral HPV infection is common in smokers and non-smokers and it is an important cause of oropharyngeal cancer in both groups.[Smith ; Gillison] Although a higher proportion of individuals with HPV-positive compared with HPV-negative tumors are non-smokers or neither smoke nor drink alcohol, many with HPV-positive disease have a history of alcohol and tobacco use. This finding underscores that HPV-associated malignant disease not only arises in people who do not smoke or drink alcohol but also occurs in people with the traditional risk factors of tobacco and alcohol use.

We have included this information with citation in the Background section.

Ref) 

Shiboski CH, Schmidt BL, Jordan RC. Tongue and tonsil carcinoma: increasing trends in the U.S. population ages 20–44 years. Cancer 2005; 103: 1843–49

Marur S, D'Souza G, Westra WH, Forastiere AA. (2010) HPV-associated head and neck cancer: a virus-related cancer epidemic. Lancet Oncol. 11:781-789

Smith EM, Ritchie JM, Summersgill KF, et al. Age, sexual behavior and human papillomavirus infection in oral cavity and oropharyngeal cancers. Int J Cancer 2004; 108: 766–72

Gillison ML, D’Souza G, Westra W, et al. Distinct risk factor profi les for human papillomavirus type 16-positive and human papillomavirus type 16-negative head and neck cancers. J Natl Cancer Inst 2008; 100: 407–20

2) “Since 1970, HPV associated head and neck cancer is increasing more rapidly in men. The average age of diagnosis is 55 to 60 years, which is about 10 years younger than HPV-associated head and neck cancer”. Clarify HPV and non-HPV cohorts

Response:

Thank you for suggestion,

The sentence corrected to “Since 1970, HPV associated head and neck cancer is increasing more rapidly in men. The average age of diagnosis is 55 to 60 years, which is about 10 years younger than HPV-unrelated head and neck cancer”

We have corrected this sentence in the manuscript.

3) For patients with distant metastasis, please consider including treatments received and tumor characteristics as a separate cohort in “Patients’ characteristics and surgical parameters” section

Response:

Thank you for your thoughtful comment.

Following your suggestion, a brief characteristic of cohort with distant metastasis included in “Risk factors of distant metastasis” section was further reinforced and moved to the “patient’s characteristics and surgical parameters” section as follows,

Of 12 (15.7%) patients who experienced a distant failure, two patients had both locoregional and distant failure with all lung metastasis. The median time to distant metastases was 32 months (range, 7–96 months). Two patients showed early distant metastasis (within 1 year). Sites of distant metastasis included lung (n = 10, 83.3%), liver (n = 1, 8.3%), and brain (n = 1, 8.3%). Ten (83.3%) patients had undergone treatment for their distant metastasis, including six patients who received chemotherapy alone, two patients who received chemoradiation, and two patients who received both surgery and chemotherapy. Surgical intervention included wedge resections for pulmonary metastases in two patients. 4 patients (33.3%) had smoking history, as same, 4 patients had alcohol history. 

For margin status, 6 patients (50%) showed close margin (1mm< <5mm) and 6 patients showed negative margin (5mm <). An infiltrative margin type was observed in 9 patients (75%) and 3 patients (25%) showed expanding margin type. 7 patients (58.3%) had lympho-vascular invasion and 1 patient (8.3%) had perineural invasion. The mean depth of invasion was 15.16 mm, and ENE was observed in 4 patients (25%).

We have included this in [Patients’ characteristics and surgical parameters] section.

4) Please cite any past papers looking at margin type and their conclusions

Response:

Thank you for suggestion.

Authors could not find studies that analyzed the margin type (infiltrative vs. expanding) in HPV related/non related OPC cancer. However, studies on the margin type of malignancy in other sites such as thyroid, colorectal could be found. Kim et al. reported clinical outcomes of papillary thyroid carcinoma according to tumor margin type. Kim found infiltrative type margin was associated with significantly higher rates of tumor multiplicity, extrathyroidal extension, coexistence of lymphocytic thyroiditis, and metastasis to lateral neck lymph nodes than the expanding type margin.[Kim] Garcia-Solano et al. published a paper on pathologic features such as tumor budding and invasive margin type in colorectal adenocarcinoma. He found that the infiltrative margin type was a worse prognostic factor for 5-year survival than the expanding type in serrated adenocarcinoma.[Garcia-Solano]

Based on these studies, the infiltrative margin type may act as a predictor of poor oncologic outcome, and it is consistent with result of this study. 

We have included this in discussion section

Ref)

Kim et, al. Tumor Margin Histology Predicts Tumor Aggressiveness in Papillary Thyroid Carcinoma: A Study of 514 Consecutive Patients. J Korean Med Sci 2011; 26: 346-351

Garcı´a-Solano et, al. Tumour budding and other prognostic pathological features at invasive margins in serrated colorectal adenocarcinoma: a comparative study with conventional carcinoma, Histopathology, 2011, 59: 1046–1056

5) Were there any patients in the study that did not receive the NCCN standard of care treatment recommendations?

Response:

The treatment protocol at this center is determined on the tumor board based on the NCCN guide line. The cohort included in this study included only the cases where observation or adjuvant therapy was performed after surgical treatment among the NCCN standard protocols. All 76 patients were treated according to the NCCN guideline.

We have included this information in the Method section. 

6) “It can be inferred that the invasiveness of infiltrative tumor margin causing distant metastasis is stronger.” This is only significant on univariate analysis so consider not including this statement

Response:

Thank you for suggestion.

Following your comment, the sentence was removed from the original manuscript.

7) “In conclusion, the current study is the largest and the most robust analysis up to date to identify”. Delete “up”

Response:

Thank you for suggestion. 

Following your comment, the sentence was corrected.

Reviewer #2: This is a retrospective review examining distant failures following primary surgical management of 76 patients with HPV-mediated tonsil cancer treated at a single institution over a 12 year period. The study aims to identify risk factors for distant failures after surgical management, which the authors note is currently lacking in the literature for patients not having undergone neoadjuvant treatment. All patients underwent primary surgical management with risk-adjusted adjuvant treatment. None had been previously treated prior to surgery. Overall survival and disease specific survival for all-comers in this cohort was excellent. Advanced T classification and positive marginal status were shown to be independent predictors of distant metastasis in this cohort by multivariate analysis. Infiltrative margin type was also associated with distant metastasis by univariate analysis only. Moreover, distant metastasis and positive marginal status were associated with significantly worsened 5-year disease-specific survival. The authors conclude that this is the largest cohort of its type, providing crucial risk stratification data and information for patient counseling in this population.

Major Points

1. The authors note that other studies have suggested that the tonsillar subsite has high rates of distant failure, but they might clarify why the tonsillar subsite was exclusively examined rather than examining both tonsillar and base of tongue subsites (or alternatively all oropharyngeal subsites). This may strengthen the study by increasing the size of the cohort and also provide additional data about the other subsites.

Response:

Thank you for your thoughtful comment.

Most of the other studies have reported recurrence or survival of the oropharynx unit, not each sub-site. Wendt reported recurrence and survival according to the sub-site of HPV-positive OPC. In his study, the primary sites consisted of tonsil (63%), base of tongue (22%), and other oropharynx (15%). In his study, in the case of tonsil, 35% of patients experienced distant failure, which is a result of examining the distant failure rate regardless of treatment modality.[Wendt] The treatment protocol at this center is determined on the tumor board based on the NCCN guideline. The cohort included in this study included only the cases where observation or adjuvant therapy was performed after surgical treatment among the NCCN standard protocols. In the case of base of tongue lesions, our center recommends neoadjuvant chemotherapy or CCRT rather than initial surgery with intend to improve the patient's swallowing functional outcome. Therefore, patients with base of tongue lesions were not included in this study.

We have included this information in the Method section. 

Ref.)

Wendt M, Hammarstedt-Nordenvall L, Zupancic M, Friesland S, Landin D, Munck-Wikland E, et al. (2021) Long-Term Survival and Recurrence in Oropharyngeal Squamous Cell Carcinoma in Relation to Subsites, HPV, and p16-Status. Cancers (Basel) 2021; 13:2553.

2. Nearly half of the patients who underwent surgery in this cohort required both adjuvant chemotherapy and radiation. It may be useful to discuss surgical selection criteria at the authors’ institution. Triple modality therapy may be unavoidable in a subset of patients with unanticipated, unresectable positive margins or unexpected extracapsular extension. However, many providers/tumor boards are likely to advocate forgoing surgery in patients who are at high risk to require triple modality therapy if subjected to surgery. While the authors report excellent overall survival comparable with other HPV-mediated cohorts, there may have been some missed opportunities to limit treatment-related morbidities in the patients undergoing triple modality treatment. How do the rates of positive margins and proportion of patients undergoing triple modality treatment in this cohort compare to those reported in other studies?

Response:

Thank you for comment.

The treatment protocol at this center is determined on the tumor board based on the NCCN guide line. In the case of advanced stage, the treatment modality is determined on the tumor board in consideration of the functional outcome after surgery, and after sufficient explanation to the patient, the treatment method is determined by respecting the patient's choice. The cohort included in this study included only the cases where observation or adjuvant therapy was performed after surgical treatment among the NCCN standard protocols. All 76 patients were agreed to initial surgical treatment. 

Regarding surgical margin, positive margin in 16 (21%), negative margin in 56 (73.6%), and close margin in 4 (5.2%) patients were found. Among patients with positive margin, 13 out of 16 (81.2%) got concurrent chemo-radiation therapy (CCRT), 2 patients (1.6%) got radiation alone therapy, and 1 patient was observed with active surveillance (patient’s refusal to adjuvant therapy). Choi et al. published a study on the prognosis according to lymph node ratio after upfront surgery in HPV-positive OPC patients, and presented positive margin and ENE as main indications for adjuvant therapy. In her study, a positive margin was found in 25 patients (27.7%) out of 90 patients, and adjuvant RT was performed in 26 patients (28.9%) and CCRT was performed in 47 patients (52.2%). Roden et al. published a study comparing outcomes of upfront surgery with adjuvant therapy or CCRT in advanced stage HPV-positive tonsil cancer patients using a national data base. Surgery was performed in 8768 patients, and positive margins were observed in 2762 (31.5%). Among cohort with positive margin, adjuvant RT was performed in 25.9% and CCRT was performed in 35.2%. These are summarized in the table below.

Ratio The present study Choi et, al. Roden et, al.

Positive margin 21% 27.7% 31.5%

Adjuvant RT 1.6% 28.9% 25.9%

Adjuvant CCRT 81.2% 52.2% 35.2%

In this study, the proportion of patients who showed a positive margin was smaller than in the other two studies, and the proportion of patients who underwent CCRT was higher than in the other two studies. This is thought to be influenced by the preference of this center tumor board.

We have included this information in the Method & Result section. 

Ref.)

Choi et, al. Importance of lymph node ratio in HPV-related oropharyngeal cancer patients treated with surgery and adjuvant treatment, PLOS ONE, 2021

Roden et, al. Triple-Modality Treatment in Patients With Advanced Stage Tonsil Cancer, 2017, cancer

3. The effect of marginal type is reported with respect to distant metastasis. Did the authors examine the relationship of infiltrative vs expanding marginal types on survival as well?

Response:

Thank you for comment.

Multivariate survival analysis considering the margin type is presented in Table 3 and it was not statistically significant (OR: 0.58, 95% CI: 0.08-4.00, p = 0.55). Following your comment, a disease specific survival curve was created for each margin type. 5-yr DSS rate of the infiltrative type was 60.5%, and the expanding type showed 83.6%. (p=0.14) 

We have included this information in the result & additional figure

Additional figure (Figure 2. Kaplan–Meier figures of disease-specific survival in patients with HPV positive tonsil cancer according to margin type)

4. Were any patients treated with immunotherapy after the diagnosis of their distant metastasis?

Response:

Distant metastasis occurred in 12 patients in this study. Two patients did not want active treatment after diagnosis of distant failure, and unfortunately, the remaining 10 patients were unable to perform immunotherapy due to economic problems (insurance coverage, etc.).

Minor Points:

1. (Page 9, paragraph 1, lines 2-3) P16 is a surrogate marker for HPV-mediated carcinoma rather than a “type.” The authors should clarify this.

Response:

Thank you for comment.

In present study, HPV status of each patient was confirmed by in situ hybridization for HPV DNA or by strong and diffuse staining for p16. There was an error in the choice of terminology. “P16” in the manuscript refers to high risk HPV 16 type. Following your comment, that sentence was corrected from “Among HPV types, p16 type is the main cause” to “Among HPV types, HPV 16 type is the main cause and it is characterized by high levels of p16 marker” (Citation is the same because it is referenced in the same paper)

We have included this information in the Background section.

2. (Page 9, paragraph 1, lines 5-6) the authors talk about HPV-associated cancer in men in the previous sentence and then reference diagnosis being 10 years earlier than HPV-associated cancer. Please clarify, since it seems like an error (are the authors trying to compare this to the age of diagnosis in women, for patients with HPV-negative carcinoma, or something else?).

Response:

Thank you for comment.

The meaning of the sentence was insufficiently made, and there was a mistake in the choice of terminology. 

The sentence corrected to “Since 1970, HPV associated head and neck cancer is increasing more rapidly in men. The average age of diagnosis is 55 to 60 years, which is about 10 years younger than HPV-unrelated head and neck cancer”

Additionally, the incidence of head and neck cancers overall has declined in recent years, consistent with the decrease in tobacco use. By contrast, incidence of HPV-associated oropharyngeal cancer seems to be rising. In particular, the prevalence is increasing rapidly in young men.[Shiboski] This effect may be attributable to changes in sexual norms (more oral sex partners or oral sex at an earlier age in recent than past generations) combined with fewer tobacco-related cancers in young cohorts, making the outcomes of HPV-positive cancers more apparent.[Marur] In addition, HPV prevalence in cervix rather than penile tissue might boost the chances of HPV infection when performing oral sex on a woman, contributing to the higher rate of HPV-associated oropharyngeal cancer in men.[Marur] In view of the importance of tobacco use in HPV related OPC, there are some study that reported oral HPV infection is common in smokers and non-smokers and it is an important cause of oropharyngeal cancer in both groups.[Smith ; Gillison] Although a higher proportion of individuals with HPV-positive compared with HPV-negative tumors are non-smokers or neither smoke nor drink alcohol, many with HPV-positive disease have a history of alcohol and tobacco use. This finding underscores that HPV-associated malignant disease not only arises in people who do not smoke or drink alcohol but also occurs in people with the traditional risk factors of tobacco and alcohol use.

Ref) 

Shiboski CH, Schmidt BL, Jordan RC. Tongue and tonsil carcinoma: increasing trends in the U.S. population ages 20–44 years. Cancer 2005; 103: 1843–49

Marur S, D'Souza G, Westra WH, Forastiere AA. (2010) HPV-associated head and neck cancer: a virus-related cancer epidemic. Lancet Oncol. 11:781-789

Smith EM, Ritchie JM, Summersgill KF, et al. Age, sexual behavior and human papillomavirus infection in oral cavity and oropharyngeal cancers. Int J Cancer 2004; 108: 766–72

Gillison ML, D’Souza G, Westra W, et al. Distinct risk factor profi les for human papillomavirus type 16-positive and human papillomavirus type 16-negative head and neck cancers. J Natl Cancer Inst 2008; 100: 407–20

We have included this information in the Background section.

3. (Page 9, paragraph 1, lines 15-16) Please correct “16 negative”

Response:

Thank you for comment. 

Following your comment, “16 negative” was corrected to “p16 negative”

We have corrected this information in the Background section.

4. (Page 13, paragraph 1, lines 3-4) The authors reference “advanced n-stage” as including N2 or N3. However, the AJCC 8 nodal classification system does not have a pathological N3 designation. Does this mean that the N-classification of these tumors was determined clinical evaluation alone or by combining available clinical and pathological data?

Response:

Thank you for comment.

There was a mistake in the choice of terminology. As you commented, nodal stage system of HPV-positive oropharyngeal cancer is consisted of N0, N1, and N2. We deleted “N3” from manuscript. Table 1. contains information of patient’s N classification and we confirmed it was configured correctly with the nodal staging system.

We have corrected this information in the result section.

5. (Page 14, paragraph 2) when reporting the univariate analysis results, it is not always clear which variables correspond to which of the percentages in parentheses. For example, which margin type had a significantly higher association with distant metastasis?

Response:

Thank you for comment.

Following your comment, variables corresponding to each rate value are indicated in the manuscript. (Advanced T stage: 53.3%, positive margin: 37.5%, infiltrative margin type: 15%)

We have included this information in the Result section.

6. The overall writing style, punctuation, and grammar should be closely examined. The authors are not consistent with indentation between various paragraphs and sections. There is also a variety of minor grammatical errors.

Response:

Thank you for suggestion.

The authors commissioned the native speaker again for the overall English proofreading.

---

## [Decision Letter · Decision Letter 1]

4 Jan 2023

PONE-D-22-18441R1Predictive factors of distant metastasis in surgically treated HPV positive tonsil cancerPLOS ONE

Dear Dr. Joo,

Thank you for submitting your manuscript to PLOS ONE. After careful consideration, we feel that it has merit but does not fully meet PLOS ONE’s publication criteria as it currently stands. Therefore, we invite you to submit a revised version of the manuscript that addresses the points raised during the review process.

We look forward to receiving your revised manuscript.

Kind regards,

Andrew Birkeland, M.D.

Academic Editor

PLOS ONE

Journal Requirements:

Additional Editor Comments:

The authors have improved their manuscript and answered the reviewer comments. There are numerous grammatical errors (improper tense, etc.) throughout the manuscript that would need to be edited, preferably by a professional editor, before acceptance.

Reviewers' comments:

Reviewer's Responses to Questions

**Comments to the Author**

1. If the authors have adequately addressed your comments raised in a previous round of review and you feel that this manuscript is now acceptable for publication, you may indicate that here to bypass the “Comments to the Author” section, enter your conflict of interest statement in the “Confidential to Editor” section, and submit your "Accept" recommendation.

Reviewer #1: All comments have been addressed

2. Is the manuscript technically sound, and do the data support the conclusions?

Reviewer #1: Yes

3. Has the statistical analysis been performed appropriately and rigorously? 

Reviewer #1: Yes

4. Have the authors made all data underlying the findings in their manuscript fully available?

Reviewer #1: No

5. Is the manuscript presented in an intelligible fashion and written in standard English?

Reviewer #1: Yes

6. Review Comments to the Author

Reviewer #1: Major and minor comments have been addressed sufficiently.

In the study population section, "Patients with images demonstrating evidence of

distant metastatic disease were re-staged and pathologic confirmation for site of distant

metastasis had tried in available cases." Please revise the wording "had tried" to be more clear.

7. PLOS authors have the option to publish the peer review history of their article (what does this mean?). If published, this will include your full peer review and any attached files.

Reviewer #1: No

---

## [Author Response · Author response to Decision Letter 1]

13 Feb 2023

Editor

Response:

Thank you for your suggestion. As the editor requested, we reviewed the current status of the papers in the reference list and the reference list has been revised as follows.

Previous Citation No. / Citation / Change

4 Marur S, D'Souza G, Westra WH, Forastiere AA. HPV-associated head and neck cancer: a virus-related cancer epidemic. The lancet oncology. 2010; 11:781-789. 

=> Deleted due to duplication with citation number 2.

17 Haberal I, Celik H, Göçmen H, Akmansu H, Yörük M, Ozeri C. Which is important in the evaluation of metastatic lymph nodes in head and neck cancer: palpation, ultrasonography, or computed tomography? Otolaryngol Head Neck Surg. 2004; 130:197-201. doi: 10.1016/j.otohns.2003.08.025 PMID: 14990916 

=> Due to the improper form of the citation, endnote file was changed to a new one.

2) The authors have improved their manuscript and answered the reviewer comments. There are numerous grammatical errors (improper tense, etc.) throughout the manuscript that would need to be edited, preferably by a professional editor, before acceptance.

Response:

Thank you for your suggestion. Additional proofreading was performed by professional native editor and the certificate for proofreading will be uploaded to the submission process.

Reviewer #1 

1) In the study population section, "Patients with images demonstrating evidence of distant metastatic disease were re-staged and pathologic confirmation for site of distant metastasis had tried in available cases." Please revise the wording "had tried" to be more clear.

Response:

Thank you for your comment. 

The sentence was revised to “Patients with images demonstrating evidence of distant metastatic disease were re-staged, and biopsy confirmation at the site of distant metastasis was attempted in anatomically accessible cases”

We have included this revision in the study population section.

---

## [Decision Letter · Decision Letter 2]

9 Mar 2023

Predictive factors of distant metastasis in surgically treated HPV positive tonsil cancer

PONE-D-22-18441R2

Dear Dr. Joo,

We’re pleased to inform you that your manuscript has been judged scientifically suitable for publication and will be formally accepted for publication once it meets all outstanding technical requirements.

Kind regards,

Andrew Birkeland, M.D.

Academic Editor

PLOS ONE

Additional Editor Comments (optional):

Reviewers' comments:

Reviewer's Responses to Questions

**Comments to the Author**

1. If the authors have adequately addressed your comments raised in a previous round of review and you feel that this manuscript is now acceptable for publication, you may indicate that here to bypass the “Comments to the Author” section, enter your conflict of interest statement in the “Confidential to Editor” section, and submit your "Accept" recommendation.

Reviewer #1: All comments have been addressed

2. Is the manuscript technically sound, and do the data support the conclusions?

Reviewer #1: Yes

3. Has the statistical analysis been performed appropriately and rigorously? 

Reviewer #1: Yes

4. Have the authors made all data underlying the findings in their manuscript fully available?

Reviewer #1: No

5. Is the manuscript presented in an intelligible fashion and written in standard English?

Reviewer #1: Yes

6. Review Comments to the Author

Reviewer #1: All comments have been adequately addressed and the paper is of adequate importance to warrant publication

7. PLOS authors have the option to publish the peer review history of their article (what does this mean?). If published, this will include your full peer review and any attached files.

Reviewer #1: No

---

## [Editor Report · Acceptance letter]

13 Mar 2023

PONE-D-22-18441R2 

Predictive factors of distant metastasis in surgically treated HPV-positive tonsil cancer 

Dear Dr. Joo:

I'm pleased to inform you that your manuscript has been deemed suitable for publication in PLOS ONE. Congratulations! Your manuscript is now with our production department. 

Kind regards, 

on behalf of

Dr. Andrew Birkeland 

Academic Editor

PLOS ONE